# Heterogeneous Parallel Implementation of Large-Scale Numerical Simulation of Saint-Venant Equations

**Yongmeng Qi [1], Qiang Li [1,\*], Zhigang Zhao [1], Jiahua Zhang [1], Lingyun Gao [2], Wu Yuan [2], Zhonghua Lu [2], Ningming Nie [2], Xiaomin Shang [1] and Shunan Tao [1]**

[1] School of Computer Science and Technology, Qingdao University, Qingdao 266071, China; 2019025931@qdu.edu.cn (Y.Q.); zgzhao@qdu.edu.cn (Z.Z.); zhangjh@radi.ac.cn (J.Z.); 2020025815@qdu.edu.cn (X.S.); 2021023842@qdu.edu.cn (S.T.)

[2] Computer Network Information Center Chinese Academy of Sciences, Beijing 100083, China; gaolingyun@cnic.cn (L.G.); yuanwu@sccas.cn (W.Y.); zhlu@sccas.cn (Z.L.); nienm@sccas.cn (N.N.)

\* Correspondence: chucklee@qdu.edu.cn; Tel.: +86-18669818931

**Abstract:** Large-scale floods are one of the major events that impact the national economy and people's livelihood every year during the flood season. Predicting the factors of flood evolution is a worldwide problem. We use the two-dimensional Saint-Venant equations as an example and for high-performance computing in modelling the flood behavior. Discretization of the two-dimensional Saint-Venant equations with initial and boundary conditions with the finite difference method in the explicit leapfrog scheme is carried out. Afterwards, we employed a large-scale heterogeneous parallel solution on the "SunRising-1" supercomputer system using MPI, OpenMP, Pthread, and OpenCL runtime libraries. On this basis, we applied communication/calculation overlapping and the local memory acceleration to optimize the performance. Finally, various performance tests of the parallel scheme are carried out from different perspectives. We have found this method is efficient and recommend this approach be used in solving systems of partial differential equations similar to the Saint-Venant equations.

**Keywords:** Saint-Venant equations; finite difference method; parallel computing; heterogeneous computing

## 1. Introduction

Reservoir dams have played a huge role in flood control, water supply, power generation, etc. [1], and are one of the important components of the hydraulic engineering system. But if a dam is damaged due to various factors, the consequences are generally catastrophic. Excessive flooding is one of the natural causes of dam failure. Floods are the most frequent natural disasters, affecting large numbers of people and agricultural lands, as well as causing casualties and damage to infrastructure. Increased runoff rates due to urbanization, prolonged rainfall, and insufficient river capacity are some of the main causes of flooding. After entering the flood season in 2020, there have been multiple rounds of heavy rainfall in southern China, resulting in severe floods in many places. As of 9 July 2020, flood disasters had affected 30.2 million people in 27 provinces (district and cities), with direct economic losses of 61.79 billion yuan. In 2021, Henan, Shanxi, and other places were also hit by heavy rainstorms. Therefore, understanding the behavior of water flow in a channel is critical for early flood disaster management and saving lives. The research direction of this paper is to study the behavior of flood through numerical simulation of surface water flow, and to perform heterogeneous parallel processing and visualization of the simulation process.

Mathematical models are scientific or engineering models constructed using mathematical logic methods and mathematical language to solve various practical problems. In hydraulic science, mathematical models are often used to simulate fluids of different phenomena, and numerical methods are used to control the fluid models. What we use in

our research is the flood wave propagation dynamics equation, which was first proposed by Saint-Venant [2], so it is also called the Saint-Venant (SV) equations. It consists of a continuity equation reflecting the law of conservation of mass and an equation of motion reflecting the law of conservation of momentum, and is widely used to predict surface flow parameters such as velocity, depth, or height. Nowadays, they are used to model flows in a wide variety of physical phenomena, such as overland flow, flooding, dam breaks, tsunami [3]. SV equation was originally only used to describe one-dimensional surface water flow, for two-dimensional, SV equation is derived from the Navier-Stokes equation [4], and two-dimensional SV equation is often referred to as the shallow water equation(SWE). Since SV equations are mathematically quasi-linear hyperbolic partial differential equations (PDE), it is difficult to obtain the analytical solution of SV equations by analytical method. Therefore, different numerical methods have been proposed to simulate surface water flow.

Long before the advent of computers, some people solved the SV equations through numerical simulation. Reinaldo and Rene [5] long ago used the explicit MacCormack time-splitting scheme to establish a mathematical model for solving two-dimensional SV equations. Two industrial applications at the time are also presented, demonstrating the validity of the model. Later, Fiedler and Ramirez [6] also used this method to simulate a discontinuous two-dimensional hydrodynamic surface flow equation (a variant of the two-dimensional SV equations) with spatially variable properties. The method was developed to model spatially variable infiltration and microtopography, and can also be used to model irrigation, tidal flat and wetland cycles, and flooding. With the development of computers and the improvement of computing performance, more and more people put the simulation process on the computer. For example, Kamboh [7] et al. also established a mathematical model with initial conditions and boundary conditions using two-dimensional Saint-Venant PDE in order to predict and simulate flood behavior. Next, the corresponding models are discretized and implemented on MATLAB using the common explicit finite difference method. The finally generated graph structure can visually see the changes of each parameter over time. Asier [8] also used the two-dimensional Saint-Venant equation to simulate precipitation or runoff events, but the method used was the finite volume method. He developed and compared the following three programming methods: sequential, multi-threaded and many-core architectures. The multi-threaded code is written using OpenMP and the many-core architecture is written using CUDA. He concluded that the performance of the GPU parallel version using CUDA is strongly affected by the size of the problem. It is also proposed that combining MPI and GPU methods can improve computational efficiency and data capacity, but this is not implemented in the paper. In the field of heterogeneous computing, Ding [9] et al. transplanted, parallelized, and accelerated the solver of the one-dimensional S-V equation based on MPI and athread library. The athread library is an accelerated thread library designed for the master-slave acceleration programming model of the SW26010 processor. They use MPI to realize the parallelization between the master cores, and use athread to accelerate the slave cores. After that, optimization methods such as SIMD (Single Instruction Multiple Data) vectorization and communication/calculation overlapping were carried out. In addition to the above work, adaptive mesh refinement (AMR) is also an important part of the algorithm performance. AMR is generally efficient and effective in treating problems with multiple spatial and temporal scales. AMR improves the quality of solution on a mesh by refining cells only in places where a high grid resolution is desired, thereby increasing the memory efficiency and computation speed [10]. Xin Zhao [11] proposed a 3D volume-of-fluid method based on the adaptive mesh refinement technique. He introduced projection methods on adaptive grids to solve the incompressible Navier–Stokes equations. In order to simulate ocean wave propagation, Michael [12] et al. proposed a method for numerical simulation of dynamically adaptive problems on adaptive triangular grids with recursive structure. They used a grid generation process based on recursive bisection of triangles along marked edges to achieve 2D dynamically adaptive discretization. Kevin and Frank [13] used a multilayer lattice Boltzmann model (LBM) to solve the 3D wind-driven shallow water flow

problems, and studied the performance of the parallel LBM for the multilayer shallow water equations on the CPU-based high performance computing environment using OpenMP. It is concluded that explicit loop control with cache optimization in LBM provides better performance than the implicit loop control on execution time, speedup, and efficiency as the number of processors increases. There are many more applications of the two-dimensional Saint-Venant equation, such as [14–17].

There are already some frameworks for solving PED using GPUs or accelerated devices. For example, Bhadke [18] et al. used CUDA to develop a 3D-CFD computing framework for the conduction process. They discretized the model into a three-dimensional grid and solved it using an alternating-direction implicit method. In summary, although many different studies have modeled the SV equation (e.g., [19,20]), and there are discussions on parallelization and performance optimization, there are few studies on large-scale Saint-Venant systems. Therefore, this research aims to build a two-dimensional simple finite difference model and use MPI, OpenMP, Pthread, and OpenCL for heterogeneous large-scale processing. In this work, we first introduce the governing equation and calculation method of SV equations. Then we introduce the basic implementation of our heterogeneous massively parallel computing. Finally, the parallel strategy is optimized and the performance is tested.

## 2. Governing Equations and Numerical Method

The governing equations used in our research are as follows:

$$\frac{\partial z}{\partial t} + \frac{\partial(zu)}{\partial x} + \frac{\partial(zv)}{\partial y} = 0 \tag{1}$$

$$\frac{\partial(zu)}{\partial t} + \frac{\partial\left(zu^2 + \frac{gz^2}{2}\right)}{\partial x} + \frac{\partial(zuv)}{\partial y} = gz\left(S_{0x} - S_{fx}\right) \tag{2}$$

$$\frac{\partial(zv)}{\partial t} + \frac{\partial(zuv)}{\partial x} + \frac{\partial\left(zv^2 + \frac{gz^2}{2}\right)}{\partial y} = gz\left(S_{0x} - S_{fx}\right) \tag{3}$$

Equation (1) is derived from the conservation of mass, and Equations (2) and (3) are derived from the conservation of momentum in the $x$ and $y$ directions, respectively. Among them, $z$ refers to the elevation (depth or height) of the water flow in the open channel, $u$ and $v$ are the water velocity in the $x$ and $y$ directions respectively, $t$ is the time, g is the acceleration of gravity. $S_0$ and $S_f$ refer to the water surface gradient and frictional resistance. In order to use this system of equations more conveniently in the computer, we need to use the product rule of differentiation to further simplify the system of equations. We then discretize the equations using an explicit finite difference scheme, where the time and space derivatives are respectively discretized by the following expressions:

$$\frac{\partial u}{\partial t} \approx \frac{u_{i,j}^{k+1} - u_{i,j}^{k-1}}{2\Delta t}, \frac{\partial u}{\partial x} \approx \frac{u_{i+1,j}^{k} - u_{i-1,j}^{k}}{2\Delta x}, \frac{\partial u}{\partial y} \approx \frac{u_{i,j+1}^{k} - u_{i,j-1}^{k}}{2\Delta y}$$

This format is the central difference format, in which the central difference in time is also called the leapfrog format, and its advantage is that it can enhance the stability of the calculation. In this way, the discretized equations become the following form:

$$\frac{z_{i,j}^{k+1} - z_{i,j}^{k-1}}{2\Delta t} + u_{i,j}^{k}\frac{z_{i+1,j}^{k} - z_{i-1,j}^{k}}{2\Delta x} + z_{i,j}^{k}\frac{u_{i+1,j}^{k} - u_{i-1,j}^{k}}{2\Delta x} + v_{i,j}^{k}\frac{z_{i,j+1}^{k} - z_{i,j-1}^{k}}{2\Delta y} + z_{i,j}^{k}\frac{v_{i,j+1}^{k} - v_{i,j-1}^{k}}{2\Delta y} = 0 \tag{4}$$

$$\begin{aligned}
&u_{i,j}^{k}\frac{z_{i,j}^{k+1} - z_{i,j}^{k-1}}{2\Delta t} + z_{i,j}^{k}\frac{u_{i,j}^{k+1} - u_{i,j}^{k-1}}{2\Delta t} + \left(u_{i,j}^{k}\right)^2\frac{z_{i+1,j}^{k} - z_{i-1,j}^{k}}{2\Delta x} + 2z_{i,j}^{k}u_{i,j}^{k}\frac{u_{i+1,j}^{k} - u_{i-1,j}^{k}}{2\Delta x} + gz_{i,j}^{k}\frac{z_{i+1,j}^{k} - z_{i-1,j}^{k}}{2\Delta x} + \\
&u_{i,j}^{k}v_{i,j}^{k}\frac{z_{i,j+1}^{k} - z_{i,j-1}^{k}}{2\Delta y} + z_{i,j}^{k}v_{i,j}^{k}\frac{u_{i,j+1}^{k} - u_{i,j-1}^{k}}{2\Delta y} + z_{i,j}^{k}u_{i,j}^{k}\frac{v_{i,j+1}^{k} - v_{i,j-1}^{k}}{2\Delta y} = gz_{i,j}^{k}\left(S_{0x} - S_{fx}\right)
\end{aligned} \tag{5}$$

$$v_{i,j}^k \frac{z_{i,j}^{k+1}-z_{i,j}^{k-1}}{2\Delta t} + z_{i,j}^k \frac{v_{i,j}^{k+1}-v_{i,j}^{k-1}}{2\Delta t} + \left(v_{i,j}^k\right)^2 \frac{z_{i,j+1}^k - z_{i,j-1}^k}{2\Delta y} + 2z_{i,j}^k v_{i,j}^k \frac{v_{i,j+1}^k - v_{i,j-1}^k}{2\Delta y} + gz_{i,j}^k \frac{z_{i,j+1}^k - z_{i,j-1}^k}{2\Delta y} +$$
$$u_{i,j}^k v_{i,j}^k \frac{z_{i+1,j}^k - z_{i-1,j}^k}{2\Delta x} + z_{i,j}^k v_{i,j}^k \frac{u_{i+1,j}^k - u_{i-1,j}^k}{2\Delta x} + z_{i,j}^k u_{i,j}^k \frac{v_{i+1,j}^k - v_{i-1,j}^k}{2\Delta x} = gz_{i,j}^k \left(S_{0y} - S_{fy}\right) \tag{6}$$

Then, after simply shifting the term and removing the denominator, the three variables $z$, $u$, and $v$ can be solved iteratively. As shown in Figure 1, each time an iteration value is calculated, the results of the previous two iterations are used, which is also a feature of the leapfrog format. Because of this, the stability of the calculation process is strengthened.

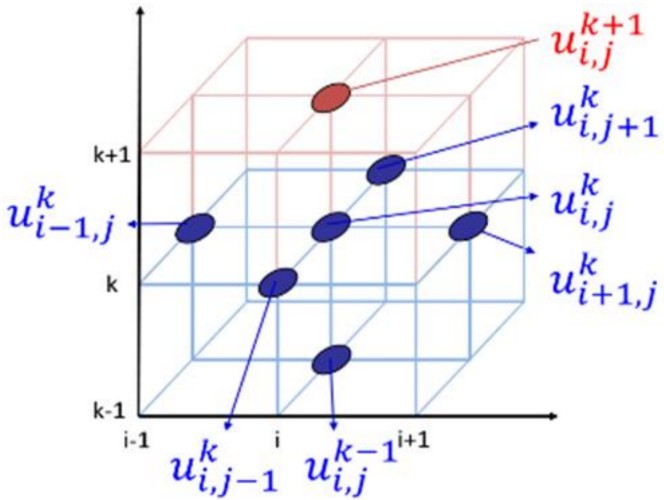

**Figure 1.** The value used by the iterative process.

It is also because of the solution value range shown in Figure 1 that the boundary of the grid will be exceeded when calculating the edge value. At this time, we need to introduce boundary conditions and add a circle of ghost cells around the standard grid. Balzano [21] gives a general review of existing wetting and drying (WD) methods. He gave expressions for boundary conditions at moving boundaries, which can be solved on fixed grids, adaptive grids and moving grids based on natural coordinates. Moreover, a number of solutions or models for dealing with boundary problems based on this method are introduced. Finally, a discussion on implicit finite-difference models is carried out. Heniche [22] et al. also gave specific boundary conditions based on the WD model. They proposed two kinds of boundary conditions: solid boundary and open boundary. Specific conditions must be imposed when dealing with solid boundaries, while open boundaries are used to specify flow regimes. The basic idea of the boundary conditions we use is to bounce incoming particles toward the boundary back into the fluid [23]. In order to achieve this boundary condition, it is only necessary to make the boundary value equal to the edge value.

After that, showing the initial conditions, we consider using a rectangular area with no bottom stress and wind stress. The water surface at each location is stationary and has a height of 10 and the flow is zero, i.e., $z$ = 10 m, $u$ = 0 m/s, $v$ = 0 m/s. Then a flood wave with a maximum height of 1 m is generated at the entrance of the water to simulate the flow behavior of fluid in a single channel, through the formula:

$$z = e^{-\frac{(a-a_0)^2 + (b-b_0)^2}{k^2}}$$

In this formula, $a_0$ and $b_0$ are the location of the highest flood wave in the x and y directions respectively, where $a_0$ = 25, $b_0$ = 1 is taken to locate the center of the entrance. $k$ is the initial height of the water wave, where $k$ = 10 m. As for $a$ and $b$, they are the coordinates

of each grid node on the coordinate axis. This will get the highest value when $a = a_0$, $b = b_0$. The initial state of a small-scale simulation process is shown in Figure 2.

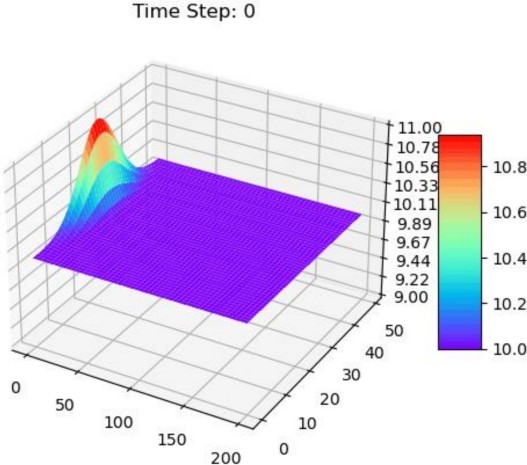

**Figure 2.** Schematic of a small-scale initial state.

## 3. Heterogeneous Implementation

First of all, our heterogeneous implementation works together through the CPU and accelerator. The CPU also participates in the calculation process while being responsible for communication, while the accelerator is only responsible for calculation. After that, we used MPI, OpenMP, Pthread, and OpenCL runtime libraries. Among them, MPI is a parallel program interface based on multiple processes with good performance, which is used in this paper for point-to-point communication between nodes. Both OpenMP and Pthread are thread-parallel interfaces. OpenMP is a portable API that is very convenient to use because it does not bind the program to a pre-set thread. But because of this, we cannot use OpenMP to manage specific threads individually. So, we introduced Pthread. The Pthread API handles most of the behavior required by threads. These behaviors include creating and terminating threads, waiting for threads to complete, and managing interactions between threads. Combining the advantages and disadvantages between the two, we use OpenMP when performing simple thread parallelism, and use Pthread when more complex thread operations are required. OpenCL is the first open, free standard for general-purpose parallel programming for heterogeneous systems. It is widely used in parallel processors such as multi-core CPUs, GPUs [24–26], accelerators, etc. We are here to process the accelerator and take care of some of the computing tasks, through OpenCL.

A heterogeneous parallel framework can be used across multiple platforms, but different clusters are equipped with different equipment, and the program will be changed accordingly. This experiment is performed on the domestic advanced computing system "SunRising-1". The specific experimental environment is shown in Table 1.

**Table 1.** SunRising-1 experimental environment.

| Hard/Software Environment | Name | Details/Version |
|---|---|---|
| Hard environment (single node) | CPU | 32-core domestic × 86 processor × 1 |
| | RAM | 16 GB DDR4 × 8 |
| | Acceleration device | Domestic GPGPU accelerator × 4, 16 GB HBM2 VRAM, bandwidth 1 TB/s |
| | Network | InfiniBand HDR network, Fat-tree topology, 200 Gbps |

**Table 1.** *Cont.*

| Hard/Software Environment | Name | Details/Version |
|---|---|---|
| Software environment | MPI | Openmpi 4.0.4 |
| | gcc/g++ | 4.8.5 |
| | OpenMP | 3.1 |
| | Pthread | NPTL 2.17 |
| | OpenCL | Platform: AMD Accelerated Parallel Processing |
| | | Driver version: 2982.0 |
| | | OpenCL Standard: OpenCL 2.0 |

Figure 3 shows the schematic of our heterogeneous parallel framework. We first initialize MPI and use MPI_Type_contiguous() to create an MPI datatype by replicating an existing datatype (for example, MPI_INT, MPI_DOUBLE, etc.). These replications are created into contiguous locations, resulting in a contiguous datatype created. The created datatype is then committed using MPI_Type_commit(), before it can be used for communication. After entering the process parallelism, routinely initialize OpenCL, including obtaining the platform, device, creating context, etc. The next step is to implement the initial value conditions described in Section 2, and we call this process "initialization". The cluster used in our experiment is equipped with 32 CPU cores and four accelerators for a single node, so for the initialization process, we enable four threads to simultaneously call OpenCL to start the kernel function. This process does not involve complicated operations, so OpenMP is used. Thread parallelism ends after the kernel function finishes running. After that the calculation process starts. In this process, both the CPU and the accelerator participate in the calculation, so we divided two thread groups, one of which contains four threads to perform operations similar to the previous process to start the kernel function, and the other is to enable the remaining 28 threads participate in the computation. The grouping and waiting of threads are involved here, so Pthread are used. After an iterative calculation is over, because of the existence of ghost cells, communication is required next. Because of the large scale of computation, a node may need to exchange data bidirectionally with four adjacent nodes (maybe 2 or 3 times for nodes at the edge). We use four MPI_Sendrecv() functions to implement this communication operation. MPI_Sendrecv() combines sending a message to a destination and receiving a message from another process into one call. Figure 4 shows how MPI_Sendrecv() in the four directions exchanges data. For the intermediate nodes, each node must send data to the surrounding adjacent nodes, and also receive data from the surrounding adjacent nodes, which can be easily implemented by calling MPI_Sendrecv().

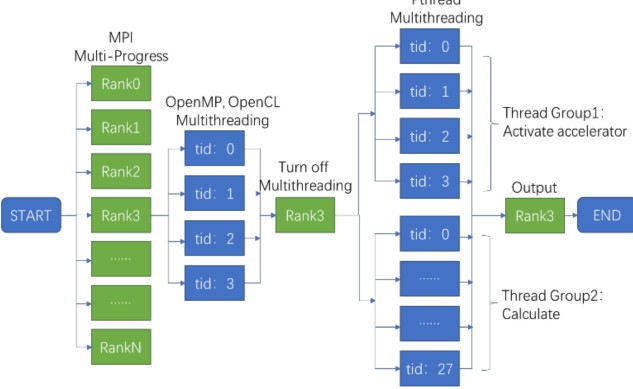

**Figure 3.** Schematic of the heterogeneous parallel framework.

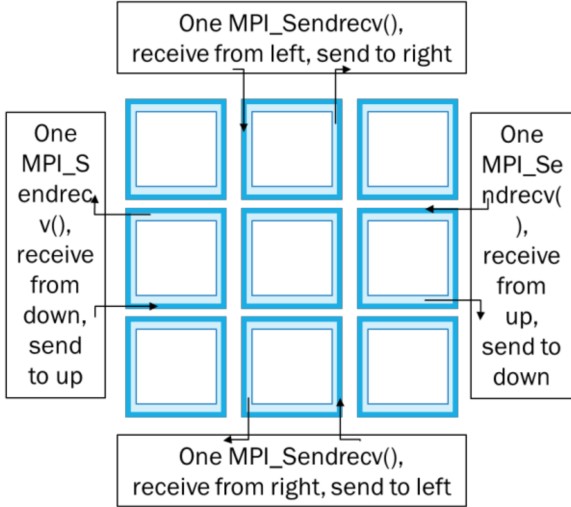

**Figure 4.** Schematic of data exchange.

When dealing with nodes at edges, we introduce MPI_PROC_NULL, a constant that represents a dummy MPI process rank. This allows all MPI processes to issue the same calls regardless of their position.

After the communication ends, a complete iteration is over, and after that, the iterative loop continues until the set time is reached. Then we output the result and perform post-processing to generate a picture, such as the visualization of the output of the z value shown in Figure 5 (Only the parts with numerical changes are displayed). In this way, you can intuitively see the change of the water level in the river channel with time.

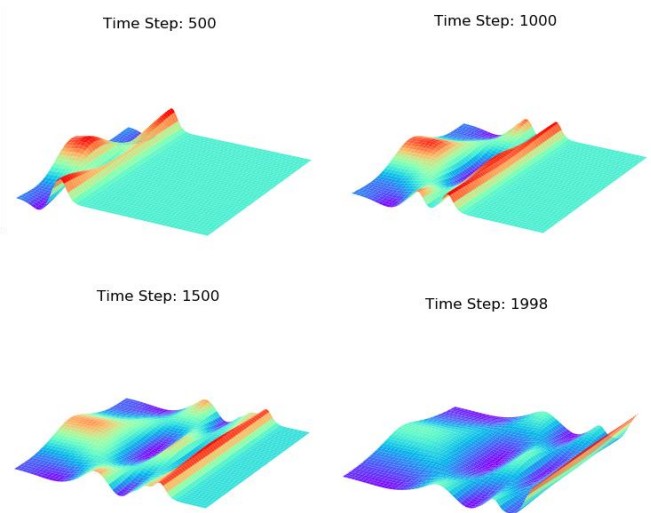

**Figure 5.** Simulation of water surface elevation at different time steps.

## 4. Performance Optimization and Testing

In this section, we will optimize the current parallel strategy and test its performance.

### 4.1. Overlap Computation and Communication

Scaling MPI applications on large high performance computing systems requires efficient communication patterns. Whenever possible, the application needs to overlap communication and computation to hide the communication latency. For the current parallel model, we consider implementing an efficient ghost cell exchange mechanism using non-blocking, peer-to-peer communication (mainly the MPI_Isend() and MPI_Irecv() functions), and domain decomposition of the grid.

First, we must confirm how to decompose the domain, that is, to confirm which part of the value to be calculated does not involve MPI communication. The grid we use is composed of real values inside and a dummy value of an outer circle of ghost cells, where real values are computed and dummy values are communicated. It can be seen from Figure 1 that the value of the ghost cell is only used when calculating the value of the outermost circle of the actual value. Therefore, as shown in Figure 6, we can come up with a solution for the domain decomposition. We call the outermost circle of the actual value the "halo cell", and the part of the actual value that removes the halo is called the "inner filed".

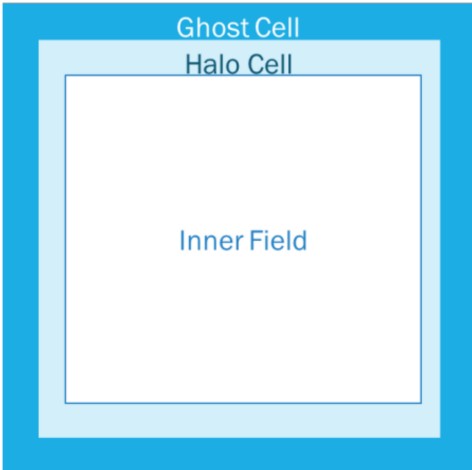

**Figure 6.** Schematic of domain decomposition.

The general iterative scheme is summarized as follows:

1. Copy data of the ghost cells to send buffers;
2. Ghost cell exchange with Isend/Irecv calls;
3. Compute (part 1): Update the inner field of the domain;
4. Call MPI_Waitall();
5. Copy data from receive buffers to the ghost cells;
6. Compute (part 2): Update the halo cells;
7. Repeat.

After completing the calculation and communication overlap, we performed a comparison test with the previous version, as shown in Table 2. The calculation amount of a single node is 200 iterations, and the grid size in each node is 2564 × 4100.

**Table 2.** Comparison test before and after overlap.

| Version | Calculating Time (s) | Sync Time (s) | Total Time (s) |
|---------|---------------------|---------------|----------------|
| Before | 240.61 | 6 | 246.61 |
| After | 189 | 5 | 194 |

In this test, we performed a total of 200 iterations, and the time obtained after the average of five times was tested. The total time simply refers to the time to execute 200 iterations, excluding the previous initialization and subsequent output time. The sync time is the time to wait with MPI_Barrier() before the end of each iteration. It can be seen that by overlapping communication and calculation, the communication process is hidden in the calculation process, which can greatly reduce the time consumption, and even when the calculation amount is large enough, the communication time can be completely ignored.

Afterwards, we conducted extended tests on this overlapped version. We tested the parallel performance of the program by continuously increasing the number of processes while keeping the amount of computation allocated by each process basically the

same. The test results are shown in the Figure 7. This test uses four nodes (processes) as the benchmark, and these four nodes are arranged in a $2 \times 2$ manner. The number of iterations and grid size are the same as before. It can be seen that the overlapping of communication computing can not only shorten the computing time but also maintain a good parallel efficiency.

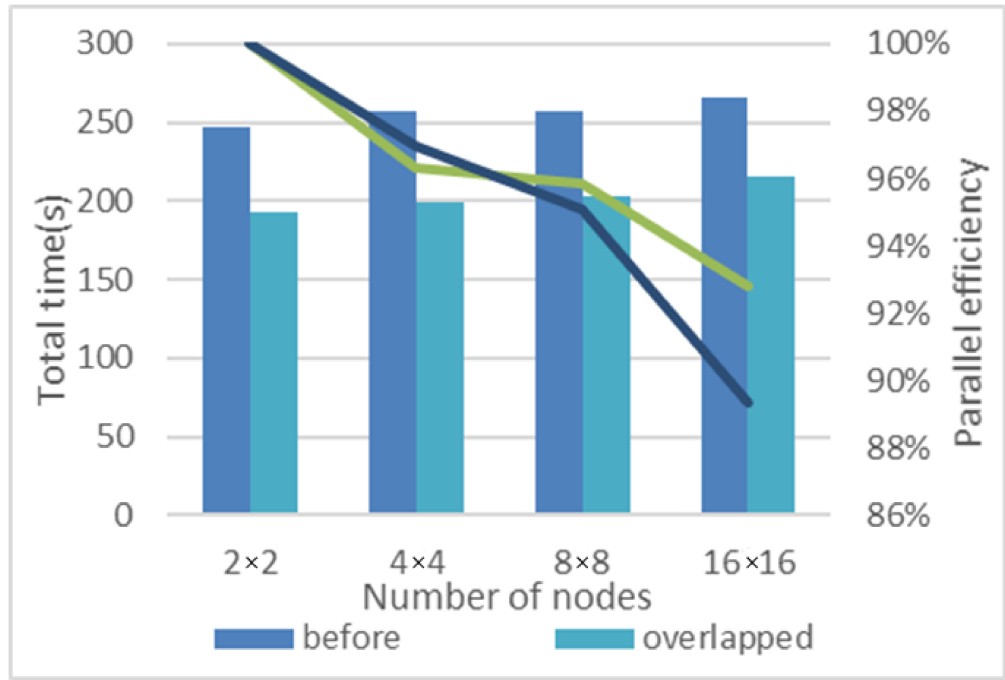

**Figure 7.** Extended test performance graph for overlapped version.

*4.2. Work Group Optimization*

First of all, a few definitions are briefly described. Each OpenCL device has one or more compute units, and each compute unit is composed of one or more processing elements. The basic unit of executing a kernel in OpenCL is called a work-item, and a collection of several work-items is called a work-group. A work-group executes on a single compute unit. The work-items in a given work-group execute concurrently on the processing elements of a single compute unit. There are two ways to specify the number of work-groups. In the explicit model a programmer defines the total number of work-items to execute in parallel and also how the work-items are divided among work-groups. In the implicit model, a programmer specifies only the total number of work-items to execute in parallel, and the division into work-groups is managed by the OpenCL implementation. We used the implicit model before, and the optimization method is to use the explicit model. In order to choose an appropriate work-group size, we conducted tests and the results are shown in the Figure 8. Finally, the default maximum work-group size is different for different devices. For example, the device limit we use is 256, which is the same as most devices. At this point, in the kernel code, add __attribute__((amdgpu_flat_work_group_size(<min>,<max>))) after the kernel function so that the kernel can be launched on when the working group is greater than 256. It can be seen from the test results that the best results can be achieved when the work-group size is 256.

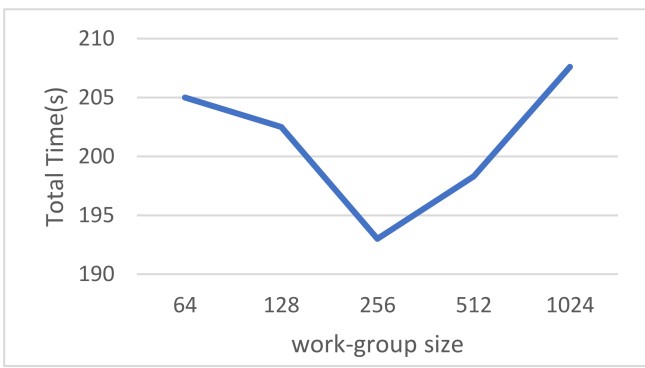

**Figure 8.** Tests for different workgroup sizes.

### 4.3. Using Local Memory

Using Local Memory is related to OpenCL's memory model. The Local Memory is the memory belonging to a certain computing unit. The host cannot see or operate on this part of the memory. This area allows all processing elements inside the computing unit to read and write, and these processing elements can share it. This also means that this is a memory area associated with a workgroup and can only be accessed by work items in that workgroup. Local Memory is the smallest unit that can be shared in the OpenCL memory structure, so making full use of Local Memory is a deep and very effective optimization method.

After fully understanding the OpenCL memory structure and the importance of using Local Memory, we start from the storage and calculation methods of the current parallel scheme, and continue to use the topology in the previous section to design the Local Memory usage scheme of this program. When this program uses the OpenCL device for calculation, it transfers the custom structure array of three iterations to the Global Memory. The first two iterations store the data that has been calculated before, and the third iteration is to store the data that will be calculated. Here, resmat (result matrix) is used to represent the array of the three iterations, resmat[0–2] represents the three iterations respectively, resmat[0] represents the first iteration, and so on to obtain the representation of the other two iterations. By analyzing the calculation formula after the simple deformation of Equations (4)–(6), the frequency of use of resmat[1] has reached 64 times, resmat[0] has 5 times, and resmat[2] also has 5 times. This means that to complete an iterative calculation, the Global Memory needs to be accessed at least 74 times in a single device, and frequent reading and writing of the Global Memory is bound to affect the performance of the program. Therefore, the optimization idea for this program is to store the data in resmat[1], which has the highest frequency of accessing the Global Memory, into the Local Memory of each work group according to the topology.

There is also a prerequisite for Local Memory optimization, that is, the size of Local Memory is limited, and the increase in read rate is achieved by sacrificing capacity. In order to ensure that the envisaged solution can be executed, we obtain the device information through the clGetDeviceInfo function, and by setting the cl_device_info parameter to CL_DEVICE_LOCAL_MEM_SIZE, the size of the Local Memory area can be obtained. In the experimental environment used by this program, the size of the Local Memory is 65,536 bytes (B), which is 64 kilobytes (KB). It can be seen that this capacity is very limited, so it is necessary to control the size of the data passed into the Local Memory.

After there are hardware limitations, go back to the Local Memory usage scheme of this program. In order to use a piece of continuous data to facilitate writing to Local Memory and the limitation of Local Memory space, only resmat[1] is considered here, and resmat[0] and resmat[2] are no longer considered. Since the topology in the previous section is used, and grid nodes and work items are in one-to-one correspondence, a work group only undertakes the computing task of a row of 256 grid nodes. When calculating this part of the task, in addition to the data of the corresponding position in resmat[1], the data of

the four adjacent positions of each grid node will be used, which can be seen in Figure 1. Therefore, processing similar to ghost cell is required, that is, the grid data of $3 \times 258$ scale is transferred to Local Memory. Each grid node consists of three double-type variables z, u, and v. Each double-type occupies 8 bytes in the system environment. Therefore, the space occupied by grid data of $3 \times 258$ scale is 18,576 B, which satisfies the limitations of OpenCL devices.

After the scheme is designed, it is the specific implementation. First, a new parameter with the __local qualifier must be added to the definition of the kernel function. This parameter is usually a pointer to a memory space, and then clSetKernelArg is used in the main function to set the parameter. At the same time, the specified size must be the size allocated by the __local parameter, and the parameter value must be NULL, because the host cannot access the Local Memory, so the data in the Global Memory can only be copied to the Local Memory in the kernel function. A set of index functions provided by OpenCL is used to determine the position of a work item in its own work group and the position in the global, so that the data in this position in the Global Memory can be copied to the corresponding position in the Local Memory. Finally, after the above processing methods, the original resmat[1] accesses the Global Memory 64 times into a single Global Memory access and 64 local memory accesses. The test results of the final program are shown in Figure 9.

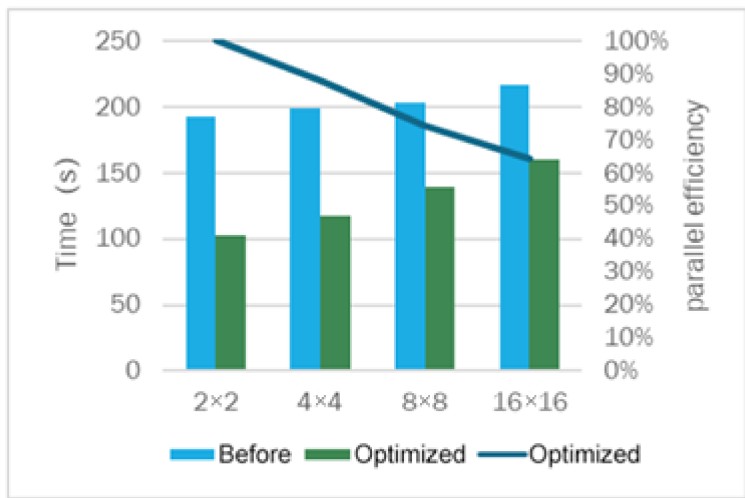

**Figure 9.** Extended test performance graph for overlapped version.

It can be seen that the running time is greatly reduced after using Local Memory, especially in the case of $2 \times 2$ nodes, the running time is optimized to 1/2 of the original. At the same time, the rate of decline in parallel efficiency has slightly increased. This is because the calculation time has been optimized, and there will be no major changes as the number of computing nodes increases. However, the original synchronization time, thread opening, and other times remain basically unchanged compared with those before optimization, and will also change with the increase of computing nodes. As the number of computing nodes increases, the proportion of computing time in the total time becomes lower, which affects the parallel efficiency, so the increase in the speed of parallel efficiency decline is within reasonable expectations.

After that, we further tested the optimization scheme. In the current scheme, the local memory usage size is 18,576 B, and a large part of the space is still unused compared to the limit of 65,536 B. In this solution, the size of the working group determines the amount of Local Memory usage. Therefore, we combined two optimization methods for testing. We started with 128 work items and increased in units of 128 to test the performance of the program under different work group sizes and using different sizes of Local Memory. The results are shown in Table 3.

**Table 3.** Comparison of test results for different workgroup sizes.

| Workgroup Size | Use Local Memory Size (B) | Test Results (s) |
|---|---|---|
| 128 | 9360 | 84.36 |
| 256 | 18,576 | 83.85 |
| 384 | 27,792 | 84.27 |
| 512 | 37,008 | 84.64 |
| 640 | 46,224 | 84.16 |
| 768 | 55,440 | 84.30 |
| 896 | 64,656 | 84.30 |
| 1024 | 73,872 | Error |

As can be seen from the table, in the case of 1024 work items, an error is reported because the size of the Local Memory used exceeds the limit of 65,536 B. In addition, the rest of the results are not very different. In OpenCL, logically all threads are parallel, but in reality, not all threads can execute at the same time from a hardware perspective, but multiple thread groups are scheduled through the hardware's own scheduling algorithm. The thread group is the smallest execution unit that is scheduled in the acceleration device defined by each hardware manufacturer. For example, on the NVIDIA CUDA platform, this thread group is called warp and consists of 32 threads, and on the AMD platform and in this lab environment, they are called wavefronts and consist of 64 threads. How many such thread groups can be executed at the same time is determined by the number or size of the Local Memory, cache, registers, and SIMD (Single Instruction, Multiple Data) instruction set of the computing unit. Therefore, under large-scale computing tasks, the total number of tasks executed at the same time is roughly the same. Therefore, it can be concluded that the size of the work group has little effect on the final result when the computing scale is large and the computing tasks are close to saturation.

Finally, we also conduct further tests on the NVIDIA platform to verify the conclusions, and the test results are shown in Table 4. This test was carried out on a single-node NVIDIA GeForce RTX 3090 platform. Due to the memory limitation of the platform, the grid node size of this test was $32 \times 1536$, and 100,000 iterations were calculated and the time was counted. It can be seen that the size of the workgroup does not affect the final result on the NVIDIA platform. Here, the NVIDIA 3090 graphics card has been tested to find that its Local Memory limit is 49,152 B, and when the workgroup size is 640 and the Local Memory size is 46,224 B, the reason for still reporting an error is that the CL_INVALID_WORK_GROUP_SIZE error in OpenCL is triggered, that is the maximum workgroup size of the OpenCL device is exceeded.

**Table 4.** Comparison test results of different workgroup sizes under the NVIDIA platform.

| Workgroup Size | Use Local Memory Size (B) | Test Results (s) |
|---|---|---|
| 128 | 9360 | 37.11 |
| 256 | 18,576 | 37.37 |
| 384 | 27,792 | 37.75 |
| 512 | 37,008 | 36.33 |
| 640 | 46,224 | Error |

*4.4. Heterogeneous Parallel Performance Testing*

First, let us take a look at the performance of heterogeneous parallelism without any optimization. In the charts in this section, the bar charts represent time and the line charts represent parallel efficiency. Only weak scaling tests are performed here, as shown in Figure 10. Because the performance of the weak extension test is not good and the parallel efficiency declines too fast, this paper does not perform multiple tests and strong extension tests on this version. After analysis, it is found that a small calculation scale is used here, as mentioned above, such problems are greatly affected by the problem size. Then we scaled

up and performed the optimizations mentioned in the previous section. Figures 11 and 12 are the results of the weak and strong scaling test after optimization.

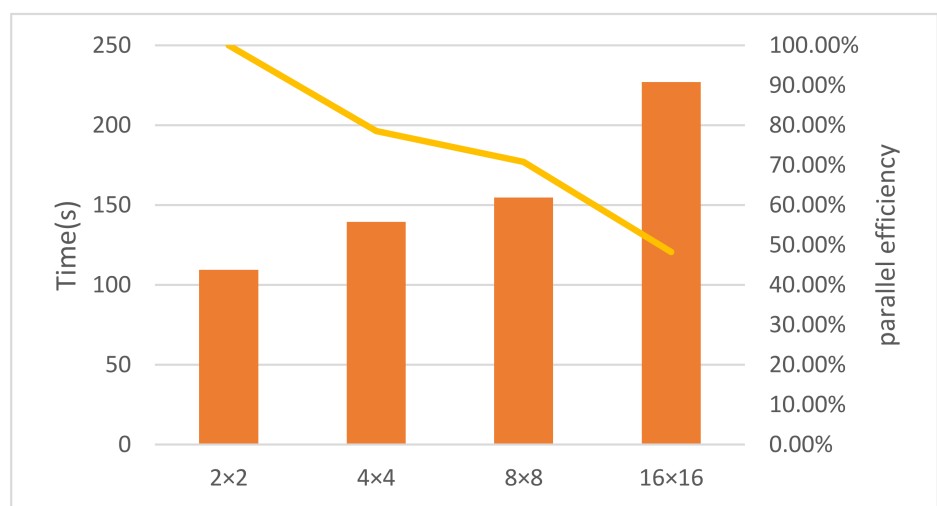

**Figure 10.** Unoptimized heterogeneous parallel weak scaling test.

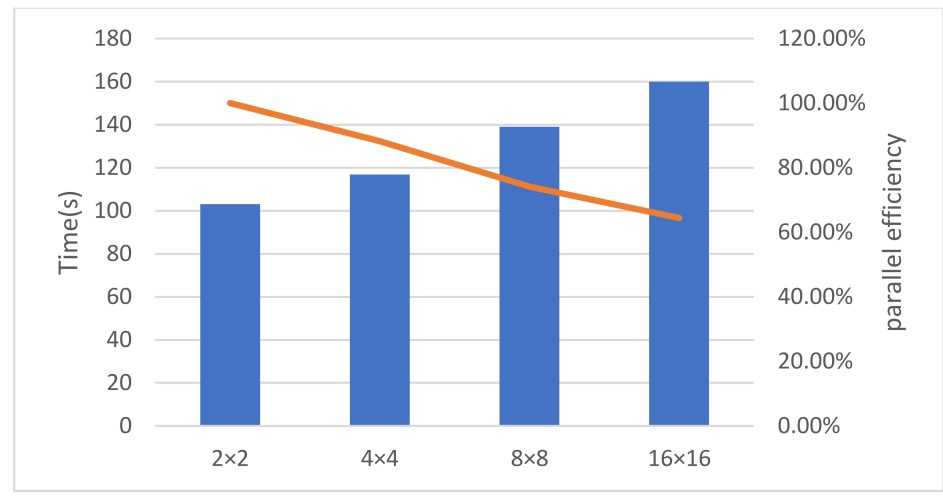

**Figure 11.** Optimized heterogeneous parallel weak scaling test.

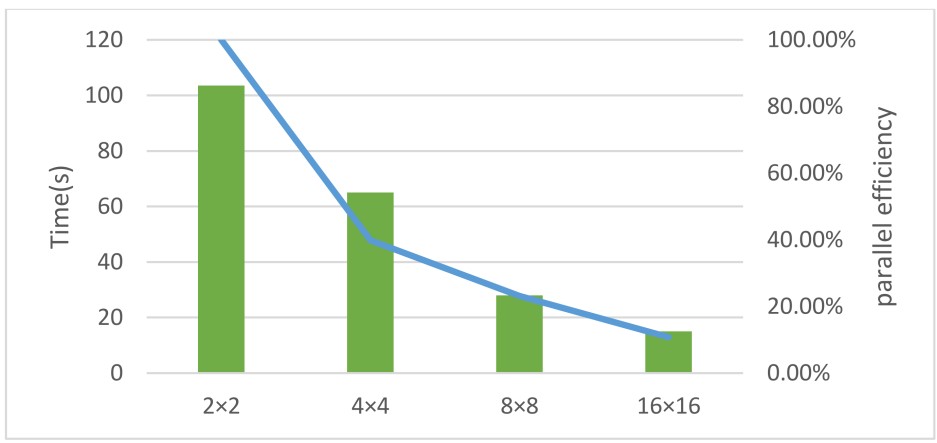

**Figure 12.** Optimized heterogeneous parallel strong scaling test.

These tests were performed on the same scale as the previous overlap optimization. It can be seen that this version of the weak scaling test can still have good parallel efficiency when the calculation time is greatly reduced. The performance of the strong scaling test is not good, because the calculation time is optimized and decreases with the number of nodes, while the synchronization time increases with the increase of computing nodes. This leads to a lower proportion of computing time in the total time, which affects the parallel efficiency. So, the increase in the rate of parallel efficiency decline is reasonably expected.

After that, the total scale is expanded to obtain better parallel efficiency. This time, the scale under $2 \times 2$ nodes is 10,244 $\times$ 16,388, and the scale under $16 \times 16$ nodes is 1284 $\times$ 2052. The test results are shown in Figure 13, it can be seen that the parallel efficiency is improved after the total scale is enlarged. In order to further obtain better parallel efficiency, the calculation scale can continue to be scaled up, but it does not make practical sense to do so.

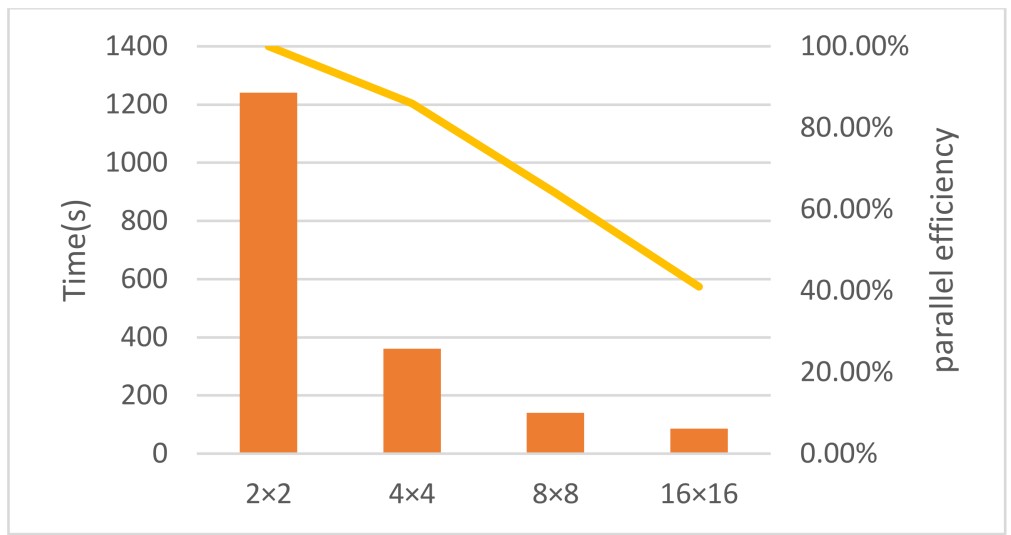

**Figure 13.** Optimized heterogeneous parallel strong scaling test. In the first section, we mentioned that Bhadke et al. also proposed a computing framework for solving PDE using GPU. However, according to our calculation method of parallel efficiency, the parallel efficiency calculated according to the speedup results provided by them is not ideal. For example, the speedup ratio calculated based on the grid size of 1000 ($10 \times 10 \times 10$) in their article, the speedup ratio also increases with the increase of the grid size, but the parallel efficiency is only below 30%.

### 4.5. Computational Time Comparison Tests

In this section, we conduct various performance comparison tests for CPU and accelerator respectively.

### 4.5.1. CPU Single-Core and Multi-Core Comparison

The first is to test the acceleration performance of the CPU using Pthread compared to the single-core serial mode. In our program, Pthread is used to launch 28 threads to be responsible for some calculations, and its acceleration performance is shown in the Figure 14. The horizontal axis is the calculation scale. We first fix the length to expand the width, and then fix the width to expand the length. It can be seen that no matter what the expansion is, the speedup is stable at around 3. Therefore, using Pthread can not only achieve more complex thread operations, but also obtain stable acceleration effects.

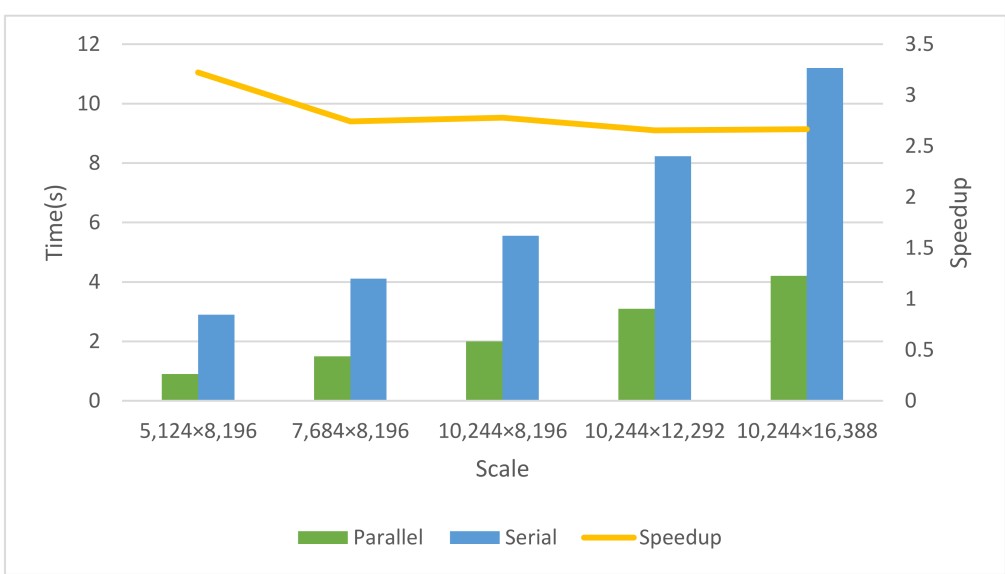

**Figure 14.** Single-core CPU vs multi-core CPU comparison chart.

### 4.5.2. CPU and Accelerators Comparison

We then compared CPU and accelerators that perform computations simultaneously. Because the cluster used in our experiment is equipped with 32 CPU cores and 4 accelerators for a single node, we did two sets of tests, one is to compare a single 28-thread CPU with a single accelerator under the same scale, and the other is to compare four accelerators with a single 28-thread CPU under the same scale. Figures 15 and 16 are the result graphs of the above two tests. It can be seen that the acceleration effect of the accelerator is very obvious, which is why a good parallel solution must use GPU or accelerator. Heterogeneous computing can make full use of the performance of CPUs and acceleration devices, reflecting that heterogeneous computing will be the future development direction of parallel computing.

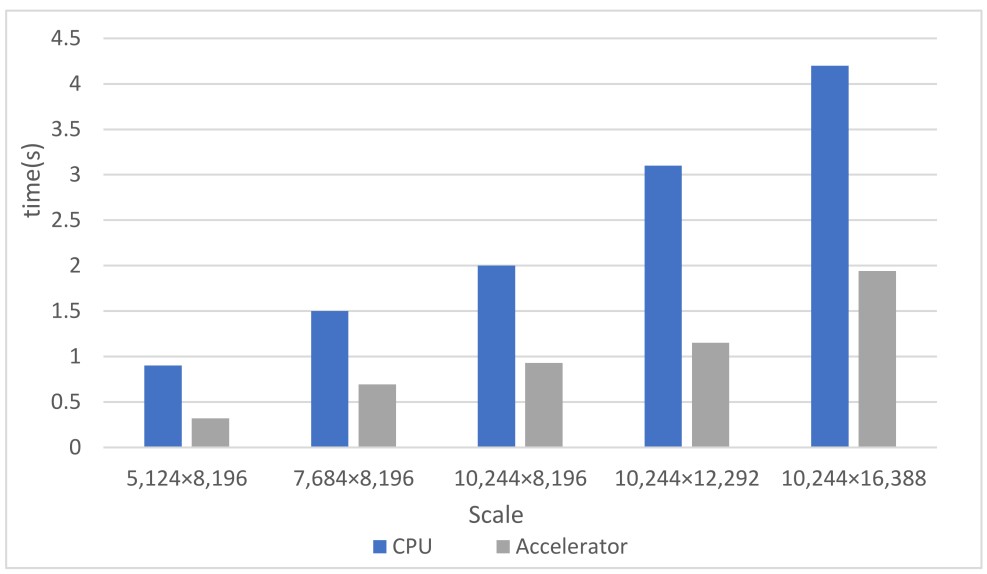

**Figure 15.** Single CPU vs. single accelerator comparison chart.

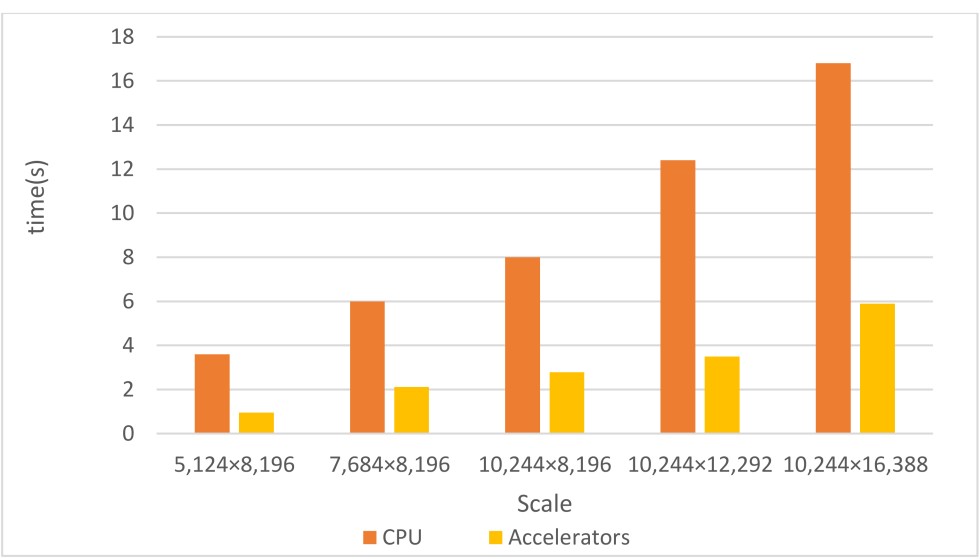

**Figure 16.** Single CPU vs. 4 accelerators comparison chart.

## 5. Conclusions

In this work, two-dimensional Saint-Venant equations were implemented by using the leapfrog-style finite difference method for the purpose of studying flood behavior. Large-scale heterogeneous parallel solution is implemented using MPI, OpenMP, Pthread, and OpenCL runtime libraries. The heterogeneous strategy is optimized by overlapping communication calculation, setting up work groups, etc., and performance tests from various perspectives are also carried out. Finally, a large-scale heterogeneous parallel solution for 2D SV equations with good performance is obtained. In the future, we hope that our work can become a large-scale heterogeneous parallel solution framework capable of solving SV equations -like PDE. For example, the heat conduction equation as an important PDE should also be able to use our computational framework. It describes how the temperature in a region changes with time, and like the Saint-Venant equation, it is difficult to obtain analytical solutions, and numerical methods are usually used to obtain numerical solutions. There are also some other problems that may also be able to use our computational framework, such as phase transitions, elasticity, electrical potential, etc. The purpose of our current research is to propose a massively heterogeneous parallel framework based on Saint-Venant's equations in domestic advanced computing systems. Therefore, a simple model is used to facilitate heterogeneous parallel implementation. In the future work, we will consider more complex and more types of models to simulate various actual situations (such as floods, tsunamis, dam failures, etc.).

**Author Contributions:** Conceptualization, Q.L. and N.N.; methodology, Q.L. and N.N.; software, Y.Q.; validation, L.G., W.Y. and Z.L.; formal analysis, Q.L.; investigation, Y.Q.; resources, Q.L.; data curation, Y.Q. and X.S.; writing—original draft preparation, Y.Q.; writing—review and editing, Q.L., Z.Z. and J.Z.; visualization, Y.Q. and S.T.; supervision, Q.L.; project administration, N.N. All authors have read and agreed to the published version of the manuscript.

**Funding:** This research was funded by the National Key R&D Program of China (Grant No. 2020YFB1709501), GHFund A (No. 20210701) and the Shandong Province Natural Science Foundation (Grant No. ZR201910310143).

**Institutional Review Board Statement:** Not applicable.

**Informed Consent Statement:** Not applicable.

**Conflicts of Interest:** The authors declare no conflict of interest.

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
