# Peer review of "Heterogeneous Parallel Implementation of Large-Scale Numerical Simulation of Saint-Venant Equations"

_applsci, doi:10.3390/app12115671_

Round 1

Reviewer 1 Report

The authors present a heterogeneous, parallel implementation of the shallow water equations (SWEs).
The language is mostly OK, but shows some weaknesses.
Although the topic is very relevant in terms of, e.g. flooding forecasts and simulation, the article lacks essentials, which are:
- the SWEs are only considered in their very basic form, abstaining from all application-relevant features (e.g. drying,wetting), which render the simulations much more challenging
- the performance optimizations are rather standard and are, generally speaking, applied to one specific problem on a standard Cartesian grid -- which has been investigated for years already and discussed in depth in literature. Thus there is only limited novelty here.
- performance results are not entirely reproducible due to lacking parameters
A more detailed list of comments is given below. I can therefore not recommend a publication.

Detailed comments:
- Sec. 1: Related work is discussed, although the topic on adaptive mesh refinement is entirely spared out -- which is however an important building block for algorithmic performance. Besides, details on the mentioned heterogeneous implementation by Asier are not given, which would be interesting in a one-liner.
- Sec. 2, P. 4: "In order to achieve this boundary condition, it is only necessary to make the boundary value equal to the edge value" -> This sounds to me like a standard outflow condition, not necessarily "bounce back"
- Sec. 2, P. 1: Description of the flood wave remains vague: "about 1m" is not scientific, please give exact numbers for z, a0, b0, k to allow for scientific reproducibility of results!
- Sec. 2: The overall description of the discretization is clear. Yet, the considered setup is a very simple one, excluding many things that render actual flooding calculations challenging (e.g., boundary treatment/wetting, ground parameterization). This can have tremendous impact on the algorithms and vectorization/parallelization concepts. The authors should comment on this, at least in the outlook, on in which sense their results are relevant for realistic setups!
- Table 1: "domestic x86" or "Domestic GPGPU" should be extended. Which processors are meant? How does their hardware architecture differ from other (potentially more widely used) hardware? etc.
- Sec. 4, 4.1: The overlapping of computation and communication is standard nowadays. The run reported in Table 2 is not comprehensive -- how many grid cells were used? What is "sync. time"?
- Sec. 4, 4.1: "This test has 2*2 four nodes" -> not clear!
- Sec. 4, 4.1: "Authors should discuss the results and how they can be interpreted from the perspective of previous studies and of the working hypotheses. The findings and their implications should be discussed in the broadest context possible. Future research directions may also be highlighted."
  -> this sentence does not make sense, probably copy-paste from paper template?
- Sec. 4, 4.2: "Finally, the default maximum work-group size is different for different devices. For example, the device limit we use is 256, which is the same as most devices."
  -> not clear; on which devices did the authors carry out the evaluations to find other than 256 sizes to be optimal?
- Fig 9: contains Chinese legend and English legend. Please remove the former!
- Sec. 4: Only weak scaling experiments are presented. Weak scaling is important, but strong scaling for representative domain sizes would be important for the actual final application developers to have an idea on how fast to solve their problem, given a big amount of computational resources.

Reviewer 2 Report

This manuscript reports a heterogeneous parallel model that is used to simulation large-scale floods. My comments are:

1 – Although the model is designed for flood modeling, this manuscript is mainly about the parallelization schemes, not the floods. In the Introduction section, the authors could provide more description on the history and the state-of-the-art parallelization methods used in flood modeling, including their strength, weakness and the reasons behind. The authors need to show that the methods they developed is novel and is of great significance in this field.

2 – Line 124: The description of the test problem needs a separate section.

3 – Line 136: A typo for “Heterogeneous”.

4 – Line 263: What determines the device limit? Does the optimal work group size always equal the device limit? If yes, what is the point of optimizing? If not, why?

5 – Figure 9: Please remove the Chinese.

6 – Section 4.4: This section is generally poorly written. The authors did not explain and analyze the results. In 4.4.2, the authors even did not describe the figures (Line 393). More description and analysis should be added and the significance of the findings should be explicitly summarized.

Reviewer 3 Report

Does this paper aim to solve the 2D SV equations numerically or introduce a mixed parallel computing method? It is confusing. 

for the numerical methods, 2nd order central scheme is used, I didn't see any novelty. Have you tried any high-order schemes? or tried any complex domains?

for the parallel computing method, it uses a combination of OpenMPI and OpenMP, why not OpenMPI only? have you compared the efficiency?

Although OpenCL and CUDA were also tested in this paper, these are relatively mature parallel computing technologies, and these cannot be published in journals as new methods. 

Round 2

Reviewer 1 Report

The authors have addressed few comments. I can still not recommend the article for publication:

  • Related work is still not sufficient, it only features few more articles from the Asian communities, but there is so much international work on efficient flooding simulations, which is not taken into account, just to name a few: "Dynamically adaptive simulations with minimal memory requirement—solving the shallow water equations using Sierpinski curves","Multilayer shallow water flow using lattice Boltzmann method with high performance computing".
  • The author claims that no papers on other boundary conditions could be found. Yet, there are many works, in particular on drying/wetting. A simple google scholar search delivers amongst others: "Evaluation of methods for numerical simulation of wetting and drying in shallow water flow models" (cited over 200 times!), "a two-dimensional finite-element drying-wetting shallow water model for rivers" (also cited over 200 times), just as starting points! Besides, it is not only about boundary conditions: the author mentions in the introduction "Increased runoff rates due to urbanization, prolonged rainfall and insufficient river capacity are some of the main causes of flooding". This is all true, but it requires more modeling of cities, percolation, etc., all of which are not subject of the paper and complicate the simulation.
  • Weak and strong scaling is now included but the performance numbers are not convincing. For the proposed problem, weak scaling should be perfect beyond 4 or 8 nodes, there is no reason why there should be performance degradations, at least not for the proposed halo-based domain decomposition. And if there are, the authors should explain them.
  • The authors claims that the CPU architecture is confidential. While I am not about to judge on this fact, some quantities have been reported such as clock speed and bandwidth which are sufficient for this very simple flow model to evaluate, e.g. with a roofline model or similar approaches, whether the sequential performance of the code is adequate. This is not done, therefore the results are not necessarily transparent with regard to actual hardware usage, i.e. is hardware really exploited by the present code. The reader would need to dig out this from resolution, variables on the grid and the time in seconds from the scalability plots which is more than cumbersome.
  • The scalability plots are not well explained. Are the bars /lines time /efficiency or vice versa? This should be included in the graphics itself. Fig 13: it is suspicious, that a speedup of more than 3(!) is observed when going from 1->2, which should by the way result in a parallel efficiency of more than 100%. But this is not given in the figure. The results do not appear trustful to me here.

Reviewer 2 Report

The quality of the manuscript has been improved after revision. I recommend an "accept" in current form.

Author Response

Thank you for accepting our paper. According to other reviewers, we have further revised the paper. If you are interested, you can read it again and give criticisms and corrections.

Reviewer 3 Report

  1. the study of the computational framework can not be made on a specific platform, the author must provide the comparison of efficiency if another method was used.
  2. Specify the availability of this computational framework for other PDE systems

Author Response

This manuscript is a resubmission of an earlier submission. The following is a list of the peer review reports and author responses from that submission.